# BLIP-3-VIDEO: YOU ONLY NEED 32 TOKENS TO REPRESENT A VIDEO EVEN IN VLMS

## ABSTRACT

We present BLIP-3-Video, a multimodal language model for videos, particularly designed to efficiently capture temporal information over multiple frames. BLIP-3-Video takes advantage of the 'temporal encoder' in addition to the conventional visual tokenizer, which maps a sequence of tokens over multiple frames into a compact set of visual tokens. This enables BLIP-3-Video to use much fewer visual tokens than its competing models (e.g., 32 vs. 4608 tokens). We explore different types of temporal encoders, including learnable spatio-temporal pooling as well as sequential models like Token Turing Machines. We experimentally confirm that BLIP-3-Video obtains video question-answering accuracies comparable to much larger state-of-the-art models (e.g., 34B), while being much smaller (i.e., 4B) and more efficient by using fewer visual tokens.

## 1 INTRODUCTION

Large Vision-Language Models (VLMs), benefiting from large-scale image-text training, have been dominating the field of computer vision. Recently, open-source VLMs are also obtaining strong results (Xue et al., 2024), despite having much smaller size than the commercial models (e.g., 4B vs. Trillions).

Further, in addition to such VLMs trained with images, VLMs for videos are becoming increasingly popular. The key component in a VLM for videos is the temporal abstraction of tokens over multiple frames. Models like Video-ChatGPT (Maaz et al., 2024) and PLLaVA (Xu et al., 2024a) rely on a simple spatial/temporal pooling on top of image frame-level tokens to represent the entire video. Some models rely on a separate video encoder to capture temporal information in videos (Lin et al., 2023). Similarly, some models use of additional convolutional layers (or Transformer layers) over frames to reduce their representation size (e.g., Video-LLaMA (Zhang et al., 2023), Kangaroo (Liu et al., 2024)). Approaches that simply collect all the visual tokens from all the frames (e.g., MiniGPT4-video (Ataallah et al., 2024), LLaVA-NeXT (Li et al., 2024b), Tarsier (Wang et al., 2024a) and LLaVA-OneVision (Wang et al., 2024a)) also have been very popular recently, as they allow capturing all the details from the frame-level tokens. However, this often makes the number of tokens for video to be very huge. Such large number of video tokens could be critical for longer videos as the LLM computation is quadratic to the number of total tokens.

In this paper, we introduce BLIP-3-Video, which is an efficient compact vision-language model with an explicit *temporal encoder*, designed particularly for videos. BLIP-3-Video particularly focuses on incorporating a learnable 'temporal encoder' within it. We explore different types of temporal encoder, and demonstrate that the model can abstract each video into much fewer visual tokens (e.g., 16) while being successful in open-ended question-answering tasks. We include a space-time attentional pooling as

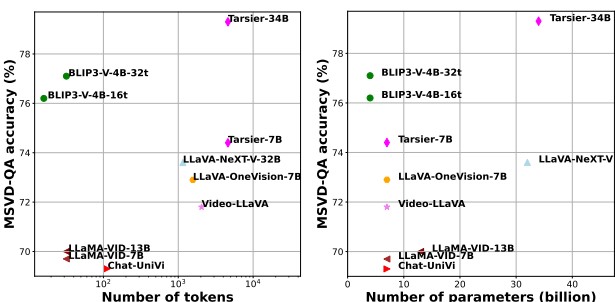

Figure 1: SOTA video VLM model comparison: (Left) Number of visual tokens vs. video-QA accuracy. (Right) Model size vs. video-QA accuracy.

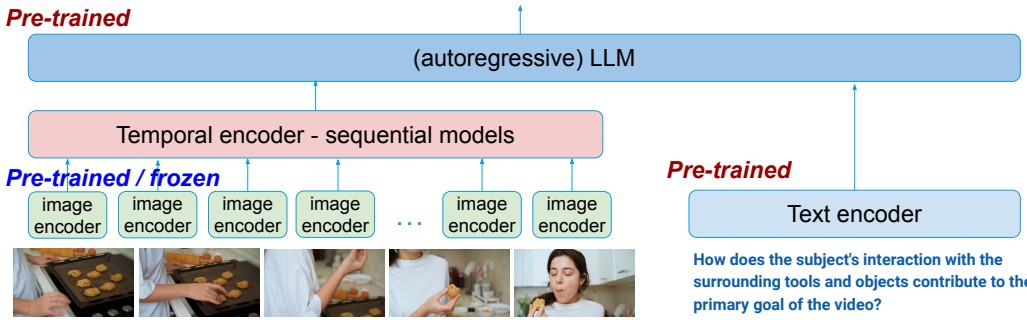

Figure 2: An illustration of the BLIP-3-Video model architecture. It has the explicit temporal encoder inserted to BLIP-3.

well as a sequential model as our temporal encoder, relying on token operations to iteratively abstract a series of frame-level tokens into a learnable memory.

There has been prior work investigating the role of pooling (Jin et al., 2024), convolutions, and cross attention layers (Zhang et al., 2023; Liu et al., 2024; Li et al., 2024c), but study on full space-time attentional pooling or sequential model to this extent has been limited in the past. Our objective in this paper is to provide a fundamental alternative to more brute-force way of collecting all the visual tokens which have been increasing popular recently. We experimentally confirm that $16 \sim 32$ video tokens abstracted by the temporal encoder is often sufficient to represent the entire video for question-answering (Figure 1).

## 2 BLIP-3-VIDEO

### 2.1 MODEL ARCHITECTURE

Our vision-language model (VLM) is an extension of the image-based VLM, BLIP-3 (Xue et al., 2024).

The model architecture is composed of the following four components: (1) the vision encoder (ViT) taking each frame input, (2) the frame-level tokenizer to reduce the number of tokens, (3) the temporal encoder to build video-level token representations, and (4) the autoregressive LLM generating output text captions based on such video tokens and text prompt tokens. Figure 2 shows an overview.

First, we apply a pretrained SigLIP as the vision encoder, designed to take one single image frame at a time. Perceiver-Resampler is then applied to map such visual tokens into $N = 128$ visual tokens per frame, independently. Once the model has such visual tokens over time (i.e., over multiple frames in the video), they are provided to an explicit 'temporal encoder'. The role of the temporal encoder is to build a video-level token representation from such sequence of image-level tokens, serving as a mapping function between a set of $N \cdot T$ image tokens to $M$ video tokens where $T$ is the number of frames and $M$ is a constant number of tokens. We explore various forms of the temporal encoder, including temporal pooling as well as sequential models, which we discuss further in the following subsection. The resulting tokens are given to the LLM together with the encoded text tokens in a prefix manner, as in many standard VLMs.

For computational efficiency, the model takes uniformly sampled 8 frames per video. As a result, in our model, ViT first maps a video into $8 * 729$ visual tokens, which is then mapped to $8 * 128$ visual tokens using Perceiver-Resampler, and then to $16 \sim 128$ video tokens using the temporal encoder.

We use Phi-3 (Abdin et al., 2024) as our LLM backbone taking such video tokens in addition to the text prompt tokens. This enables the model to take text+video as an input and generate text sentences as an output.

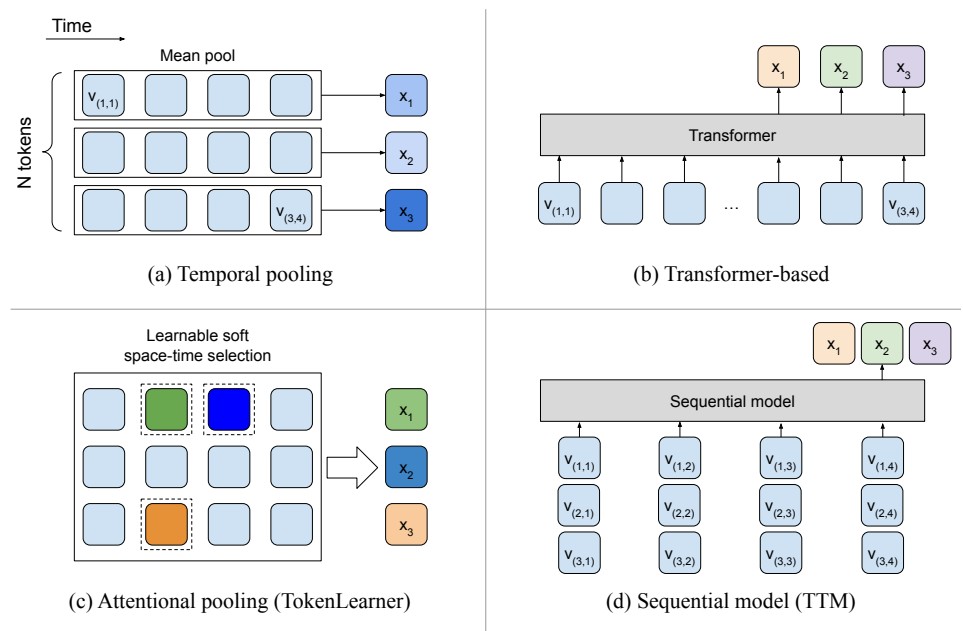

(a) Temporal pooling      (b) Transformer-based

(c) Attentional pooling (TokenLearner)      (d) Sequential model (TTM)

Figure 3: Visually comparing different types of temporal encoders we explored in our model architecture.

## 2.2 TEMPORAL ENCODERS

A temporal encoder is a function of tokens, taking $N \cdot T$ tokens as an input and returning $M$ tokens as an output: $x_{1,...,M} = f(v_{(1,1),...,(N,T)})$.

We explore different types of encoders as part of our model. The simplest form of the temporal encoder will be temporal pooling, e.g., summating per-frame tokens over time: $x_{1,...,M} = \left\{ \sum_t (v_{(i,t)}) \right\}_{i=1}^{M}$ where $N$ is always restricted to be identical to $M$, which was also used in (Maaz et al., 2024). Another possible implementation would be the use of a temporal Transformer, modeling the entire token sequence and selecting the last $m$ tokens similar to Mirasol3B (Piergiovanni et al., 2024):

$$x_{1,...,M} = \{\text{Transformer}(v)\}_{N \cdot T - M + 1}^{N \cdot T} \tag{1}$$

In addition to the straightforward temporal encoders mentioned above, we explore two important temporal encoders considering space-time nature of tokens: spatio-temporal attentional pooling and sequential models (Figure 3).

**Spatio-temporal attentional pooling:** Attentional pooling allows learnable 'soft selection' of multiple tokens given a larger set of tokens. Such attentional pooling also have been previously developed in Transformers (e.g., Perceiver (Jaegle et al., 2022) and TokenLearner (Ryoo et al., 2021)), and in earlier foundation models (e.g., CoCa (Yu et al., 2022)) for images.

In our model, we use TokenLearner (Ryoo et al., 2021), making it explicitly serve as our space-time aware temporal encoder. Unlike previous per-image-frame usage of poolings where spatial pooling and temporal pooling are applied separately (e.g., Video-ChatGPT), our temporal encoder directly takes all $N \cdot T$ tokens and 'learns' to soft-select informative tokens spatio-temporally. Here, $N$ tokens could be viewed as spatial representations of a frame and we have $T$ of them, forming a spatio-temporal representation.

Our attentional pooling in its simplest form is expressed as:

$$x_i = \alpha(V) \cdot V = \text{softmax}\left(\alpha(V^T)\right) \cdot V \tag{2}$$

where $V$ is a matrix formed by concatenating input tokens $v_{(1,1),...,(N,T)}$. The function $\alpha(\cdot)$ computes the summation weights for $V$, performing soft selection of tokens. In Perceiver, a matrix multiplication

with a latent query tokens (i.e., $|Q| = m$) have been used to implement cross attention (i.e., $\alpha(V) = Q \cdot V^T / c$). TokenLearner uses a convolution/MLP on top of $V$: $\alpha(V) = \text{MLP}_m(V^T)$, which we use in our model. This allows selecting a smaller number of tokens (e.g., $M = 32$ tokens).

We experimentally confirm that such learnable spatio-temporal attentional pooling has advantages over the conventional approach of non-learnable spatial pooling and temporal pooling, in Section 3.3.

**Sequential Model:** We also deploy Token Turing Machines (TTM) (Ryoo et al., 2023) as a temporal encoder, which is a sequential model capable of taking any number of frames to generate a video-level token representation (e.g., $M = 32$ regardless the number of frames). Our use of TTM is similar to its usage in Mirasol3B (Piergiovanni et al., 2024), except that our model uses TTM directly to encode a sequence of image tokens while Mirasol3B uses TTM to encode a sequence of low-level video tokens. We also further extend TTM by adding time-stamped positional encodings to embded the frame index of each token in the latent space. This enables the tokens in the 'memory' of TTM to preserve the temporal ordering information, which is crucial when representing complicated or long video scenes. In addition, we use TTM temporal encoder in a 'grouped' fashion, maintaining a separate memory of size $G$ for each of $N$ tokens over time. The final output from the sequence model is attentionally pooled from the final memory (whose size is $N \cdot G$).

### 2.3 TRAINING RECIPE

BLIP-3-Video follows a three-stage curriculum learning: (1) image caption pretraining, (2) video caption pretraining, and (3) video instruction tuning. In all its training we freeze the vision encoder, only training the model parameters after the vision encoder. First, we directly use the pretrained weights from BLIP-3 (Xue et al., 2024). BLIP-3 is for images and it does not contain weights for the temporal encoder, so we randomly initialize those weights.

The model is then finetuned on LLaVA-Hound-DPO's video caption data (Zhang et al., 2024b), featuring over 900k video captions. Instead of directly using the text captions provided in LLaVA-Hound-DPO, we used GPT-4 to rephrase such text captions so that they become more GPT-style captions. We also experimented with replacing the rephrased LLaVA-Hound-DPO's video caption data with a filtered version of the Mira caption dataset (Ju et al., 2024), where we excluded captions for videos longer than one minute, totaling 935k samples. LLaVA-Hound-DPO caption data performed superior to Mira on question-answering, while Mira dataset was better for the video captioning.

Finally, we tuned the model using a mix of video question-answering datasets, including VideoChat-GPT's 99k-sample video instruction tuning data (Maaz et al., 2024), along with the training splits of the MSVD-QA (Xu et al., 2017), MSRVTT-QA (Xu et al., 2017), ActivityNet-QA (Yu et al., 2019), and NExT-QA (Xiao et al., 2021) datasets, which contain 30k, 149k, 32k, and 34k samples, respectively. For the MSVD, MSRVTT, and NExT-QA training data, we used GPT-3.5 to rephrase the original single-word or single-phrase answer into a natural language sentence, providing the question in the LLM prompt context. Both open-ended and multiple-choice video QA formats are used for NExT-QA in our video instruction tuning recipe.

We trained our model with $8 \times$ H100 GPUs. For the video caption pretraining, we use the batch size of 16 per GPU, 500 warmup steps, and the learning rate of 2e-5 with the cosine decay. We trained the model for 1 epoch. The video QA sft (i.e., instruction tuning) was done with the batch size of 4 per gpu, 500 warmup steps, and the learning rate of 1e-5 with the cosine decay. We trained the model for 1 epoch in this case as well. The entire training takes around 6 hours, confirming the efficiency of our model.

## 3 EXPERIMENTS AND RESULTS

### 3.1 MODEL IMPLEMENTATION DETAILS

We share the model details with BLIP-3 4B, except that BLIP-3-Video has the temporal encoder. This model takes the video with input resolution of 384×384, using SigLIP encoder to map it to 729 tokens per frame with the channel size 1152. Perceiver-resampler is implemented with multiple cross-attention layers with the same channel dim, which is then given to the temporal encoder.

| Method | Size | #tokens | MSVD-QA | MSRVTT-QA | ActivityNet-QA | TGIF-QA |
|---|---|---|---|---|---|---|
| VideoChat (Li et al., 2023b) | 7B | 32 | 56.3 / 2.8 | 45.0 / 2.5 | - / 2.2 | 34.4 / 2.3 |
| Video-LLaMA (Zhang et al., 2023) | 7B | 32 | 51.6 / 2.5 | 29.6 / 1.8 | 12.4 / 1.1 | - / - |
| Video-ChatGPT (Maaz et al., 2024) | 7B | 264+ | 64.9 / 3.3 | 49.3 / 2.8 | 34.2 / 2.8 | 51.4 / 3.0 |
| Chat-UniVi (Jin et al., 2024) | 7B | 112 | 69.3 / 3.7 | 55.0 / 3.1 | 46.1 / 3.3 | 69.0 / 3.8 |
| LLaMA-VID (Li et al., 2024c) | 7B | 32 | 69.7 / 3.7 | 57.7 / 3.2 | 47.4 / 3.3 | - |
| LLaMA-VID (Li et al., 2024c) | 13B | 32 | 70.0 / 3.7 | 58.9 / 3.3 | 47.5 / 3.3 | |
| Video-LLaVA (Lin et al., 2023) | 7B | 2048 | 71.8 / 3.9 | 59.2 / 3.5 | 45.3 / 3.3 | 70.0 / 4.0 |
| MiniGPT4-Video (Ataallah et al., 2024) | 7B | 2880+ | 73.9 / 4.1 | 59.7 / 3.3 | 46.3 / 3.4 | 72.2 / 4.1 |
| PLLaVA (Xu et al., 2024a) | 7B | 576+ | 76.6 / 4.1 | 62.0 / 3.5 | 56.3 / 3.5 | 77.5 / 4.1 |
| SlowFast-LLaVA Xu et al. (2024b) | 7B | 3680 | 79.1 / 4.1 | 65.8 / 3.6 | 56.3 / 3.4 | 78.7 / 4.2 |
| LLaVA-Hound-DPO Zhang et al. (2024b) | 7B | 2048 | 80.7 / 4.1 | 70.2 / 3.7 | - / - | 61.4 / 3.5 |
| LLaVA-OneVision* (Wang et al., 2024a) | 7B | 1568 | 72.9 / 3.9 | 57.8 / 3.4 | 55.3 / 3.6 | 41.1 / 3.1 |
| Tarsier (Wang et al., 2024a) | 7B | 4608+ | 77.0 / 4.1 | 62.0 / 3.5 | 59.5 / 3.6 | 79.2 / 4.2 |
| Tarsier * (Wang et al., 2024a) | 7B | 4608 | 74.4 / 4.0 | 59.1 / 3.4 | 54.3 / 3.5 | - / - |
| PLLaVA (Xu et al., 2024a) | 34B | 576+ | 79.9 / 4.2 | 68.7 / 3.8 | 60.9 / 3.7 | 80.6 / 4.3 |
| LLaVA-NeXT-Video* (Li et al., 2024b) | 32B | 1152 | 73.6 / 4.0 | 56.8 / 3.4 | 58.4 / 3.6 | 73.5 / 4.1 |
| Tarsier (Wang et al., 2024a) | 34B | 4608+ | 80.3 / 4.2 | 66.4 / 3.7 | 61.6 / 3.7 | 82.5 / 4.4 |
| Tarsier * (Wang et al., 2024a) | 34B | 4608+ | 79.3 / 4.1 | 62.2 / 3.5 | 61.5 / 3.7 | - / - |
| BLIP-3-Video | **4B** | 32 | 77.1 / 4.2 | 60.0 / 3.6 | 55.7 / 3.5 | 77.1 / 4.3 |
| BLIP-3-Video | **4B** | 128 | 77.3 / 4.2 | 59.7 / 3.6 | 56.7 / 3.6 | 77.1 / 4.3 |

Table 1: Comparison against reported numbers of other models on open-ended question answering evaluation. The number of visual tokens are also reported. The numbers after '/' are answer quality scores. * indicates our evaluation using the checkpoint and inference code provided by the author, with the identical videos used in our model (8 frames of 384×384 resolution).

TokenLearner serving as the spatio-temporal attentional pooling was implemented using a MLP as the attention function. The size of its inner dim was the number of target tokens * 2. The grouped TTM serving as the sequential model temporal encoder was implemented using 4 Transformer layers (with the channel dim of 1152) as the processor module while using TokenLearners for read/write modules. Memory size was set to 128 tokens total.

The resulting $16 \sim 128$ tokens are mapped to the text embedding dimension of 3072, before given to the LLM (Phi-3).

## 3.2 PUBLIC BENCHMARKS

We conducted experiments measuring video question-answering accuracies on multiple public datasets. This includes open-ended answer generation tasks like MSVD-QA, as well as multiple choice questions like NExT-QA. We follow their standard settings in all cases.

Table 1 compare open-ended question answering accuracies of BLIP-3-Video against reported numbers of other models. We use four commonly used public datasets, MSVD-QA, MSRVTT-QA, ActivityNet-QA, and TGIF-QA, following standard VideoLLM evaluation settings. Note that our MSVD-QA and MSRVTT-QA accuracy was measured by training our model with a subset (i.e., Video-ChatGPT dataset-only) of our training data, in order to avoid the training data contamination. We are including the model size as well as the number of visual tokens in the table. We are able to observe that, despite its smaller size (i.e., 4B vs. 7B or 34B), our model is obtaining superior or comparable performance.

With the temporal encoder, BLIP-3-Video was able to retain the performance with much fewer tokens, which we discuss more in the following subsection. Our results suggest that not too many visual tokens are really necessary to be successful on these open-ended question answering benchmarks, as long as we have a carefully designed temporal encoder.

In addition, we evaluated BLIP-3-Video's ability to solve multiple choice questions (MCQ). Table 2 shows the results on NExT-QA. Due to the nature of its questions requiring understanding of multiple frames, many prior models use quie a bit of tokens. For instance, GPT-4 uses a minimum of 255 tokens per frame. It is interesting that BLIP-3-Video achieves comparable accuracy while representing the entire video with only 32 (or 128) tokens.

| Method | Size | #tokens | NExT-QA |
|---|---|---|---|
| LangRepo (Kahatapitiya et al., 2024) | 7B | 3136+ | 54.6 |
| LangRepo (Kahatapitiya et al., 2024) | 12B | 3136+ | 60.9 |
| Tarsier (Wang et al., 2024a) | 7B | 4608+ | 71.6 |
| LLoVi (Zhang et al., 2024a) | 157B | 1000s | 67.7 |
| IG-VLM (Kim et al., 2024) | 34B | 1536+ | 70.9 |
| VideoAgent (Wang et al., 2024b) | GPT-4 | 2091+ | 71.3 |
| VideoTree (Wang et al., 2024c) | GPT-4 | 3978+ | 73.5 |
| Tarsier (Wang et al., 2024a) | 34B | 4608+ | 79.2 |
| BLIP-3-Video | **4B** | **32** | 76.4 |
| BLIP-3-Video | **4B** | 128 | 77.1 |

Table 2: Comparison against reported numbers of other models on multiple choice question-answering (MCQ) benchmark.

| Encoder | MSVD-QA | TGIF-QA | ActivityNet-QA | NExT-QA |
|---|---|---|---|---|
| 1 frame | 71.49 / 4.01 | 72.74 / 4.16 | 51.83 / 3.39 | 72.79 |
| Mean pooling | 76.75 / 4.17 | 77.01 /4.30 | 55.89 / 3.53 | 76.24 |
| Transformer | 76.24 / 4.15 | 76.33 / 4.28 | 55.59 / 3.50 | 76.34 |
| Vanilla Token Turing Machine | 76.42 / 4.15 | 75.80 / 4.26 | 54.45 / 3.48 | 75.42 |
| Ours (Space-time) | 77.49 / 4.18 | 76.90 / 4.29 | 56.94 / 3.56 | 76.27 |
| Ours (Sequential) | 77.29 / 4.18 | 77.10 / 4.31 | 56.66 / 3.56 | 77.07 |

Table 3: Ablations comparing different temporal encoders: 128 tokens

## 3.3 ABLATIONS

We conducted an ablation comparing different temporal encoders. These include: (1) the base single frame model (i.e., BLIP-3 trained with videos), (2) mean pooling similar to Video-ChatGPT, and (3) transformer temporal encoder similar to Mirasol3B. We also tried the (4) vanilla Token Turing Machines, which is not the grouped version we use as our temporal encoder.

Table 3 shows the result, comparing the question-answering accuracies of different types of temporal encoders when abstracting a video into 128 tokens. We are able to observe that they all do a reasonable job, while some temporal encoders are more effective.

In addition, we compared different pooling approaches similar to the ones tried in prior works, when they are required to select a much smaller number of tokens (e.g., 32) from a large set of visual tokens. We compare our spatio-temporal attentional pooling as well as the sequential model against its alternatives, including (1) fixed-window space-time pooling and (2) learnable per-frame pooling. In particular, (2) is similar to the approach taken in LLaMA-VID (Li et al., 2024c), which independently selected a fixed number of tokens (e.g., 2) per frame. Table 4 shows the results.

Table 5 explicitly compares the impact of having smaller visual tokens. 32 visual tokens or more seem to give a reasonable video QA accuracy.

**Speed:** Reducing the number of visual tokens increases the computational efficiency of the models, as the total computation is quadratic to the number of tokens fed to the LLM. We measure the runtime of our models in the training setting for the fair comparison. Here, we report 'samples per second per GPU'. Without the temporal encoder (i.e., directly using 1024 visual tokens), the model processed 3.3 samples per second. With 16/32/128 tokens using the temporal encoder, the model was able to process 8.5 / 8.2 / 7.5 samples per second.

| Encoder | MSVD-QA |
|---|---|
| Space-time pooling (4*8) | 76.04 |
| Per-frame (4*8) | 76.78 |
| Ours (Space-time) | 77.71 |
| Ours (Sequential) | 77.11 |

Table 4: Ablations comparing different pooling strategies for 32 tokens.

| # tokens | MSVD-QA | TGIF-QA | NExT-QA |
|---|---|---|---|
| 16 tokens | 76.17 / 4.16 | 76.19 / 4.28 | 75.8 |
| 32 tokens | 77.11 / 4.17 | 77.07 / 4.30 | 76.4 |
| 128 tokens | 77.29 / 4.18 | 77.10 / 4.31 | 77.07 |
| 256 tokens | 77.67 / 4.18 | 77.35 / 4.31 | 77.06 |

Table 5: Ablations comparing different # of tokens. Ours with sequential model as a temporal encoder was used.

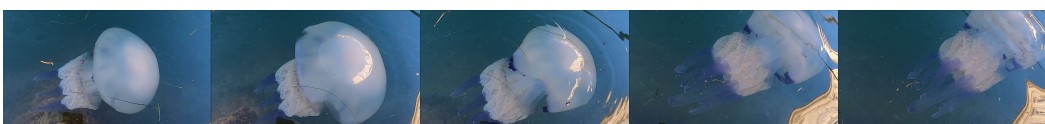

**Question**: Please describe the primary object or subject in the video, capturing their attributes, actions, positions, and movements.
**GT**: A white jellyfish, is seen floating and gently pulsating as it moves through the water. Its bell contracts and expands rhythmically, propelling it forward. The tentacles trail behind, undulating with the water's movement. Throughout the sequence, the jellyfish maintains a steady, graceful motion, drifting from left to right and slightly rotating as it progresses.
**Prediction**: A jellyfish, is seen swimming in the water. It moves in a slow, undulating motion, with its bell-shaped body contracting and expanding to propel itself forward. The jellyfish's tentacles trail behind it, creating a delicate, flowing effect. Throughout the video, the jellyfish maintains a consistent upward and forward movement, occasionally adjusting its direction slightly.

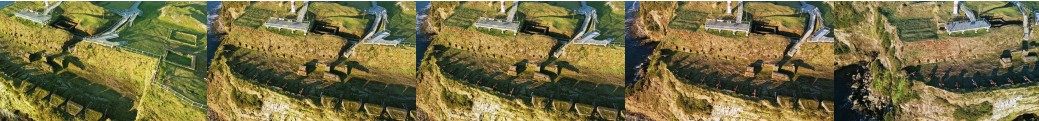

**Question**: Please provide a dense description of the video, covering the main subject, movements, style, backgrounds, and cameras.
**GT**: The video provides a breathtaking aerial view of a historic castle perched on the edge of a steep cliff. As the drone camera glides over the scene, viewers can appreciate the castle's ancient stone walls and the surrounding lush greenery, which contrasts sharply with the rugged cliff faces. The castle, a relic of medieval architecture, is shown in various states of preservation, with some parts reduced to ruins while others stand robustly. The video captures the castle's strategic location, offering panoramic views of the surrounding landscape and the churning waters below.
**Prediction**: The video provides an aerial view of a historic fortification perched on a rugged cliff overlooking the ocean. The fort, characterized by its robust stone walls and strategic positioning, includes various defensive structures such as cannons and barricades. As the video progresses, the camera captures the changing light of the setting sun, casting long shadows and highlighting the textures of the stone and the surrounding landscape. The fort's layout, including pathways and small buildings, is revealed in greater detail, showcasing its historical significance and architectural complexity.

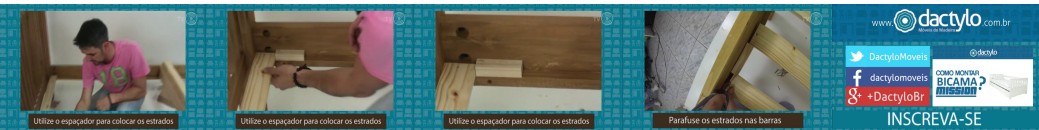

**Question**: Please provide a dense description of the video, covering the main subject, movements, style, backgrounds, and cameras.
**GT**: A step-by-step guide on assembling a wooden bed frame, featuring a person methodically working through the construction process. Initially, the person is seen preparing the wooden planks and arranging the necessary tools. As the video progresses, the individual meticulously attaches and secures the slats to the frame, ensuring stability and precision. The final scenes depict the person tightening the last screws and reviewing the completed structure, demonstrating a sense of accomplishment and attention to detail.
**Prediction**: A step-by-step process of assembling a piece of furniture, specifically a bed frame. The person involved is seen working on a wooden bed frame, using various tools and components to construct the bed. The video likely serves as a tutorial, providing viewers with a visual guide on how to assemble the bed frame themselves. The person's actions are deliberate and precise, indicating a level of expertise or familiarity with the task.

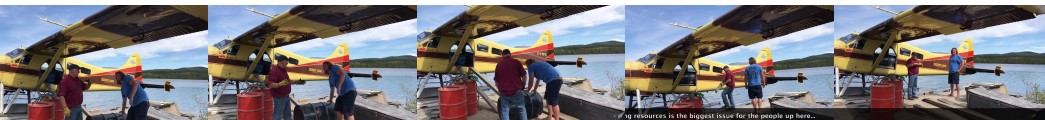

**Question**: Please provide a dense description of the video, covering the main subject, movements, style, backgrounds, and cameras.
**GT**: Two men engaged in the task of loading a barrel onto a small, yellow floatplane near a body of water. The men are dressed in casual attire suitable for outdoor work, with one wearing gloves for handling the barrel. The plane is docked at a wooden structure, possibly a jetty or makeshift dock, and the men coordinate their efforts to lift and maneuver the barrel into the cargo area of the aircraft. The surrounding environment is serene and natural, with a forested backdrop suggesting a remote or wilderness setting.
**Prediction**: A sequence of events where two men are engaged in loading a barrel onto a small, yellow seaplane parked on a wooden dock beside a serene lake. The men, dressed in casual work attire, appear to be coordinating their efforts to secure the barrel, which is large and cylindrical, onto the aircraft. The setting is tranquil, with the calm waters of the lake reflecting the clear blue sky and the lush greenery of the surrounding landscape. The video conveys a sense of teamwork and the meticulous nature of preparing an aircraft for a journey.

Figure 4: Example video captioning results on Mira dataset, formed in question-answering style.

| Method | MSVD-Cap | MSRVTT-Cap | Mira-Cap |
|---|---|---|---|
| LLaVA-OneVision-7B | 61.68 / 3.31 | 38.56 / 2.70 | 48.83 / 3.10 |
| Tarsier-7B | 63.54 / 3.41 | -/- | 40.88 / 2.88 |
| Ours | 69.50 / 3.52 | 50.45 / 2.98 | 81.76 / 4.00 |

Table 6: Video Caption Evaluation Results. We employ VideoChatGPT's LLM evaluation and report Average Accuracy / Average Score in this table

## 3.4 VIDEO CAPTIONING EVALUATION

We evaluate our model on the video captioning task by comparing it against state-of-the-art models on the test splits of MSVD-Caption and MSRVTT-Caption, as well as a custom evaluation split from the Mira dataset. For the Mira dataset, we randomly selected 6,000 samples from our full, filtered data to create the evaluation split, with the remainder used for training. We employed Video-ChatGPT's

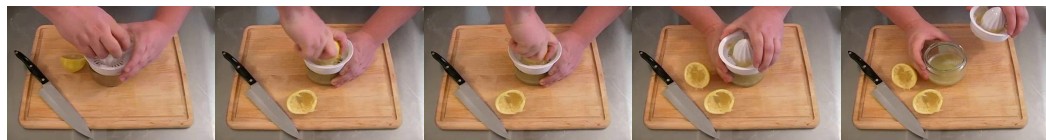

Prediction: A person is using a citrus juicer to extract juice from a lemon on a wooden cutting board. The person's hands are visible, and they are wearing a black sleeve. The lemon is placed on the cutting board, and the person squeezes it to extract the juice, which is collected in a white bowl. The process is methodical and focused, with the person ensuring to extract as much juice as possible.

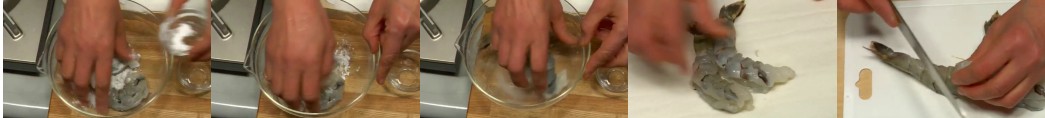

Prediction: A person is preparing shrimp for cooking. They start by placing the shrimp in a glass bowl and adding a white powdery substance, likely salt or flour. The person then mixes the shrimp and the powder together with their hands, ensuring the shrimp are evenly coated. Finally, the person transfers the shrimp onto a cutting board, ready for the next step in the cooking process.

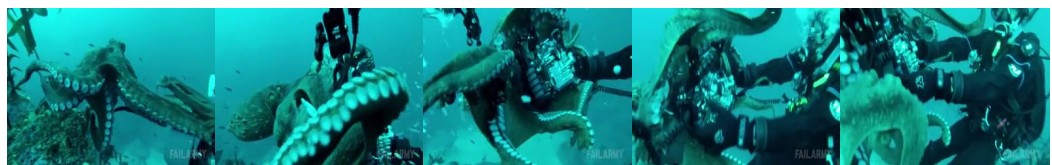

Prediction: The video depicts a dramatic encounter between a scuba diver and an octopus in the ocean. The octopus, with its tentacles spread wide, appears to be approaching the diver, who is equipped with a camera and other diving gear. The octopus's movements are fluid and deliberate, showcasing its ability to maneuver through the water with ease. The diver remains relatively stationary, observing the octopus's behavior.

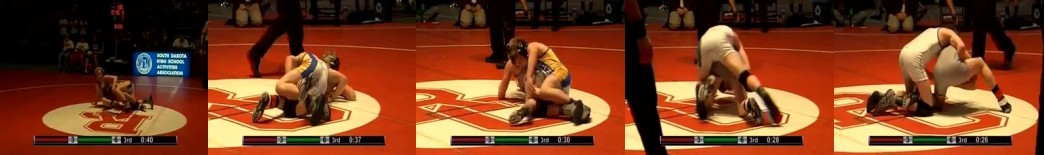

Prediction: A wrestling match between two athletes on a red mat. The wrestlers are engaged in a series of competitive maneuvers, attempting to gain control over each other. The sequence shows the wrestlers in various positions, from standing to on the ground, as they grapple and execute moves. The intensity of the match is evident through their physical exertion and strategic positioning.

Figure 5: Example video captioning results on MSVD and MSRVTT caption dataset.

LLM evaluation, specifically using GPT-3.5 to compare model-predicted captions with ground truth captions. The LLM assesses accuracy by checking if the predicted caption matches the ground truth, and assigns a score on a scale of 0 to 5 for each sample.

Table 6 demonstrates the results. All three models were provided with 8 frames per video, and consistent visual input and prompts were ensured across the models. Our BLIP-3-Video consistently outperforms LLaVA-OneVision-7B and Tarsier-7B across all three video captioning benchmarks, with particularly notable improvements on the Mira video captioning task.

We present qualitative video captioning results for the Mira dataset in Figure 4 and for the MSVD and MSRVTT datasets in Figure 5. BLIP-3-Video generates high-quality, detailed captions.

## 4 RELATED WORKS

### 4.1 IMAGE-TEXT LLMs

Among recent advances in image-text multimodal models (Li et al., 2023a; Alayrac et al., 2022; Liu et al., 2023; Dai et al., 2023; Xue et al., 2024; Laurençon et al., 2024; Deitke et al., 2024), one common strategy enable image understanding in LLM is to start with a pre-trained image encoder (e.g., ViT (Radford et al., 2021; Zhai et al., 2023)) and a pre-trained language-only LLM (Abdin et al., 2024; Bai et al., 2023; Dubey et al., 2024). The two components are connected via a vision-language connector, which is trained to project vision embeddings output from the vision encoder into "vision tokens" that can be ingested by the LLM. The vision tokens are of the same shape as language embeddings, so the image-text LLM can be trained in the same way as regular language models using the next token prediction loss. There are many design choices for the VL connector, for

example, BLIP-2 (Li et al., 2023a) chooses to use a Q-Former to extract vision tokens from the vision embeddings, Flamingo (Alayrac et al., 2022) uses "perceiver resampler" as the connector plus cross-attention layers throughout the language model, while a simpler choice is to use MLP layers to transform the vision embeddings. Image-text LLMs are usually trained with a multi-stage training strategy, including pre-training, instruction tuning, and sometimes, post-training (e.g., DPO (Rafailov et al., 2024)). In addition to simple structured image-text data such as image captioning and single-image VQA, recent works also explore free-from image-text data for model training such as interleaved image-text understanding (Laurençon et al., 2023; Awadalla et al., 2024) and multi-image VQA (Jiang et al., 2024; Li et al., 2024a).

## 4.2 VIDEO LLMS

Video LLMs extend the architecture of image-based LLMs to handle video input. Zhang et al. (2023) integrates pre-trained encoders and frozen LLMs to process multimodal input through a Video Q-Former and Audio Q-Former, generating video and audio embeddings compatible with LLM without retraining encoders. Maaz et al. (2024) adapts the CLIP visual encoder for video by incorporating temporal features and fine-tunes the model using video-instruction pairs collected by tools like BLIP-2 (Li et al., 2023a) and GRiT (Wu et al., 2022). Li et al. (2024c) generates frame-level embeddings using a visual encoder but condenses visual information into two tokens per frame. However, it does not account for temporal recency across frames. Similarly, models like Video-LLaVA (Lin et al., 2023) and LLaVa-OneVision (Li et al., 2024a) treat videos as long multi-image contexts but lack token efficiency optimization, making them computationally costly. SlowFast-LLaVA (Xu et al., 2024b) adopts a two-stream architecture—Slow and Fast pathways—to capture both spatial and temporal video semantics without extra fine-tuning. Finally, LLaVa-hound-DPO (Zhang et al., 2024b) uses Direct Preference Optimization (DPO) (Rafailov et al., 2024) with GPT-4V to annotate preference data, enhancing video question-answering performance by detecting inconsistencies or hallucinations in model responses.

## 4.3 TOKEN PRUNING

Token pruning is a widely used technique to reduce redundant and overlapping information in Vision Transformers (ViTs) and large language models (LLMs). Bolya et al. (2022) merges similar tokens within ViTs, combining redundant content while retaining task-relevant information across tasks like image, video, and audio processing. Similarly, Ren et al. (2023) employs the Temporal Aggregation Module to combine redundant consecutive video frames and the Spatial Aggregation Module to merge similar patches within each frame, reducing the number of processed tokens by up to 75%. Shen et al. (2024) focus on temporal redundancy and progressively merges tokens across neighboring clips, which reduces the number of tokens by preserving important video-level features. All these methods focus on visual token merging in ViTs, where token processing is challenging in video-based LLMs. In addition, Chen et al. (2024) improves attention efficiency in deeper layers by dynamically pruning or merging redundant image tokens based on attention scores without extra training. Shang et al. (2024) introduces adaptive token reduction through the Adaptive Important Token Selection and Token Supplement, which can be integrated into VLM models without fine-tuning. In LLMs, KV cache pruning is popular for efficient model serving, as seen in (Fu et al., 2024), which uses attention maps to progressively prune tokens and reduce the time-to-first-token (TTFT). Wan et al. (2024) extends KV cache pruning to VLMs, employing different token merging strategies to cut computational costs and support longer multimodal contexts.

## 5 CONCLUSION

We introduce BLIP-3-Video, which is an efficient, compact vision-language model for videos with 4B parameters. BLIP-3-Video incorporates a temporal encoder in its architecture, which allows the model to abstract the entire video with as few as 16 or 32 tokens. In contrast to many state-of-the-art video VLMs taking advantage of thousands of visual tokens to represent a video (e.g., 4608), BLIP-3-Video shows a competitive performance while utilizing much fewer visual tokens (e.g., 32).

**Reproducibility Statement** We build on top of the open-source BLIP-3 (XGen-MM) model and training code hosted on Huggingface and github. All the experiments were conducted with public datasets. The code and the trained model will be released together with the final version of the paper.

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
