# OpenReview forum: "BLIP-3-Video: You Only Need 32 Tokens to Represent a Video Even in VLMs"
_ICLR.cc/2025/Conference — Submitted to ICLR 2025_

### Official Review · Reviewer_7c93 · 2024-10-28

**Soundness:** 2
**Presentation:** 2
**Contribution:** 1
**Rating:** 6
**Confidence:** 4

**Summary:**

The paper introduces BLIP-3-Video, a novel multimodal language model designed for video understanding that efficiently captures temporal information across frames. A key innovation is the integration of a 'temporal encoder' that maps a sequence of tokens from multiple frames into a compact set of visual tokens, allowing BLIP-3-Video to operate with significantly fewer visual tokens compared to its competitors. The model explores various temporal encoders, including learnable spatio-temporal pooling and sequential models like Token Turing Machines. Experiments demonstrate that BLIP-3-Video achieves comparable video question-answering accuracies to much larger models, while being more efficient and smaller in size due to its reduced token usage. The paper also details the model's architecture, training recipe, and experimental results, highlighting the effectiveness of the temporal encoder in representing videos for question-answering tasks with a small number of tokens.

**Strengths:**

- **Originality**: It introduces an innovative temporal encoder that significantly reduces the number of visual tokens needed to represent videos, offering a new approach to efficiency in video understanding models.

- **Quality**: The model is thoroughly evaluated against state-of-the-art benchmarks, demonstrating competitive performance. The ablation studies are providing insightful analyses into the model's components.

- **Clarity**: The paper is well-organized, with clear explanations and visual aids that effectively convey complex information, making the technical content accessible to readers.

- **Significance**: BLIP-3-Video's efficiency in handling video data with fewer resources.

**Weaknesses:**

- **Diversity of Datasets**: The experiments primarily rely on a limited set of public benchmarks for evaluation. Expanding the evaluation to include a more diverse range of benchmarks, particularly those with varying lengths and complexities of videos, could provide a more comprehensive assessment of the model's generalizability and robustness.

- **Scalability Analysis**: While the paper demonstrates the model's efficiency, there is a lack of analysis on how the model scales with increasing video length and complexity. Future work could benefit from exploring the model's performance as it processes longer videos, which is crucial for real-world applications.

- **Comparison with State-of-the-Art**: Although comparisons are made with other models, the paper could benefit from a more detailed analysis comparing the trade-offs between BLIP-3-Video and the state-of-the-art models in terms of accuracy, computational resources, and inference time.

- **Implementation Details**: Some aspects of the model's implementation, such as the specific choices made in the architecture of the temporal encoder, could be elaborated upon with more technical depth. This additional detail would aid other researchers in understanding the design decisions and potentially replicating or improving upon them.

**Questions:**

1. **Temporal Encoder Generalization**: A smaller number of visual tokens is particularly important for understanding longer videos. However, this paper only tested on a few simple short video benchmarks. Please provide test results on video benchmarks of different lengths and scenarios, such as VideoMME, MVBench, etc.

2. **Scalability Concerns**: How does the model's performance and efficiency scale with longer videos, and have you observed any limitations in terms of the number of frames the model can effectively process?

3. **Model Interpretability**: The paper mentions the use of different types of temporal encoders. Are there any plans to provide insights into how these encoders make decisions?

4. **Comparison with Other Efficient Models**: How does BLIP-3-Video compare with other recent models that also focus on efficiency, such as those employing knowledge distillation or sparse attention mechanisms? Could the authors provide some insights into the trade-offs involved? Please provide persuasive evidence from experimental studies.

5. **Novelty**: Compared to LLaMA-VID, where is the core novelty of this paper? Although the experimental results show that 32 tokens achieved better performance on four short video benchmarks, this standard will change with different video lengths, video scenarios, and the complexity of question answering. The scalability and generalizability of this method are questionable. Perhaps a more effective mechanism for accommodating more frames and selecting key information for video question answering from a large number of visual tokens is worth exploring, rather than determining the specific numerical value of a visually overfitted token on a few benchmarks. Similar architectures have been explored enough in a series of works such as Video-LLaVA, LLaVA-VID, LLaVA-NEXT, and so on.

---

> ### Author Response · Authors · 2024-11-23
> **Response to reviewer 7c93 (1/4)**
>
> We thank the reviewer for the comments. Please find our answer to address the concerns below.
>
> 1.
> > "Diversity of Datasets: The experiments primarily rely on a limited set of public benchmarks for evaluation. Expanding the evaluation to include a more diverse range of benchmarks, particularly those with varying lengths and complexities of videos, could provide a more comprehensive assessment of the model's generalizability and robustness."
>
> Following the suggestions from the reviewers, we evaluated our model on multiple additional datasets: MVBench, TempCompass, and VideoInstruct.
>
> We find BLIP-3-Video quite competitive in all these benchmarks, particularly considering its size (3.9B) and the number of visual tokens (often less than 1/20 of the others). Notably, we performed quite a bit better than LLaMA-VID-7B, which also uses 32 visual tokens like ours (and has a larger LLM).
>
> BLIP-3-Video’s result on MVBench is as below. Notably, it is the 2nd best among the models not taking advantage of the MVBench-provided training dataset (VideoChat2-IT).
>
> | Model | # tokens | {VideoChat2-IT training} | MVBench Accuracy |
> | --- | --- | :---: | --- |
> | PLLaVA (7B) | 576+ | Y | 46.6 |
> | VideoLLaMA2 (7B) | 1152 | Y | 54.6 |
> | ST-LLM (7B) | 256 | ~Y | 54.9 |
> | PPLLaVA (7B) | 1024 | ~Y | 59.2 |
> | VideoChat2-Mistral (7B) | 96 | Y | 60.4 |
> | Kangaroo (8B) | ~10000 | Y | 61.1 |
> | Tarsier (7B) | 4608+ | ~Y | 62.6 |
> | | | | |
> | VideoChatGPT (7B) | 264+ | N | 32.7 |
> | VideoLLaMA (7B) | 32 | N | 34.1 |
> | VideoChat (7B) | 32 | N | 35.5 |
> | LLaMA-VID (7B) | 32 | N | 41.4 |
> | Video-LLaVA (7B) | 2048 | N | 43.5 |
> | mPLUG-Owl3 (8B)| n/a | N | 54.5 |
> | **BLIP-3-Video (3.9B)** | 32 | N | 54.9 |
> | LLaVA-OneVision (7B) | 3136 | N | 56.7 |
>
> Our result compared to SOTA on the TempCompass benchmark is as below.
>
> | Model | Yes/No QA | Caption matching |
> | --- | --- | --- |
> | GPT-4o | 73.66 | 80.84 |
> | Qwen2-VL-7B-Instruct | 72.77 | 77.31 |
> | Gemini-1.5-pro | 70.32 | 77.45 |
> | LLaVA-OneVision-Qwen-2-7B | 69.67 | 73.79 |
> | LLaVA-NeXT-Video-32B-Qwen | 69.38 | 76.51 |
> | InternVL2-8B | 68.24 | 77.11 |
> | **BLIP-3-Video (3.9B)** | 66.7 | 66.5 |
> | Llama-3-VILA1.5-8B | 63.64 | 68.93 |
> | LongVA-7B | 62.13 | 65.67 |
> | LLaVA-NeXT-Video-7B-DPO | 61.19 | 63.01 |
> | VideoChat2-vicuna-stage3 | 58.01 | 53.69 |
> | LLaVA-1.5-13B | 56.38 | 64.27 |
> | Video-LLaVA-7B | 56.38 | 63.34 | 63.34 |
> | Video-LLaMA-2-13B | 53.73 | 54.16 |
> | LLaMA-VID-7B-short-video | 52.96 | 56.02 |
>
> VideoInstruct benchmark evaluation also gave us similar results:
>
> | Model  | VideoInstruct accuracy |
> | --- | --- |
> | PLLaVA-34B | 3.32 |
> | SlowFast-LLaVA-34B | 3.32 |
> | VideoGPT+ | 3.28 |
> | ST-LLM-7B | 3.15 |
> | **BLIP-3-Video (3.9B)** | 3.11 |
> | VideoChat2_HD_mistral | 3.10 |
> | LITA-13B | 3.04 |
> | LLaMA-VID-13B | 2.99 |
> | VideoChat2 | 2.98 |
> | LLaMA-VID-7B | 2.89 |
> | Video-ChatGPT | 2.38 |
>
> We believe BLIP-3-Video performs very reasonably on all these benchmarks, considering its smaller size and its use of much fewer visual tokens.
>
> Also, we would like to highlight that we evaluated our model with a diverse set of datasets with different video durations. As shown in the table below, they range from very short video clips to longer clips.
>
> | Dataset  | # of videos | Average duration (sec.) |
> | --- | --- | --- |
> | TGIF-QA | 165,165 | 3 |
> | MSVD-QA | 13,157 | 10 |
> | MSRVTT-QA |  72,821 | 15 |
> | MVBench | 4,000 | 16 |
> | NExT-QA | 52,044 | 44 |
> | ActivityNet-QA | 800 | 111 |

---

> ### Author Response · Authors · 2024-11-23
> **Response to reviewer 7c93 (2/4)**
>
> 2.
> > "Scalability Analysis: While the paper demonstrates the model's efficiency, there is a lack of analysis on how the model scales with increasing video length and complexity. Future work could benefit from exploring the model's performance as it processes longer videos, which is crucial for real-world applications."
>
> The focus of this paper has been on capturing temporal information over shorter (~2 min) video segments. What we show in the paper is that our model with the temporal encoder mechanism enables efficient capturing of information in video segments (with temporal changes) compared to prior work. What we confirm is that it is a better/more efficient way to handle a video clip.
>
> Handling longer videos is an important research area, and we can offer the following insights. In order to handle long videos (e.g., movies), additional (hierarchical) LLM mechanisms similar to LLovi, LangRepo, or VideoTree would be very useful. Such framework design is orthogonal to our work; our model can serve as the VLM component within those types of frameworks, enabling extraction of better video information from local/short segments that higher-level LLM will combine for the long video modeling.
>
> 3.
> > "Comparison with State-of-the-Art: Although comparisons are made with other models, the paper could benefit from a more detailed analysis comparing the trade-offs between BLIP-3-Video and the state-of-the-art models in terms of accuracy, computational resources, and inference time."
>
> Thank you for the suggestion. In our paper, we have provided figures and tables to show them. We show (1) the trade-off between model accuracy and the number of visual tokens, which is a good proxy for inference time (i.e., more tokens directly suggest heavier compute). Figure 1 (left) as well as Tables 1 and 2 include such information. The actual inference time depends on the hardware being used, and we find the number of visual tokens to be a fair proxy for the comparison.
>
> In addition, we included (2) the trade-off between model accuracy and model size, which serves as a good proxy for the computational resources. The larger the model is, the more computational resources are required. Figure 1 (right) as well as the model sizes we specified in Tables 1 and 2 correspond to this information.

---

> ### Author Response · Authors · 2024-11-23
> **Response to reviewer 7c93 (3/4)**
>
> Q1.
> > "Temporal Encoder Generalization: A smaller number of visual tokens is particularly important for understanding longer videos. However, this paper only tested on a few simple short video benchmarks. Please provide test results on video benchmarks of different lengths and scenarios, such as VideoMME, MVBench, etc."
>
> Following the suggestion from the reviewer, we evaluated BLIP-3-Video on MVBench. Please find the table in the above post (Answer 1). As discussed above, we find BLIP-3-Video quite competitive in this benchmark. While having a much smaller model size of 3.9B (i.e., using less computational resources) and utilizing much less visual tokens (i.e., faster inference), it performed comparably to the SOTA models. Among the models not trained with MVBench-provided training set, it ranked the 2nd, only next to the LLaVA-OneVision using 100x more visual tokens. Compared to other models using 32 tokens like LLaMA-VID-7B, it performed significantly better despite using a smaller LLM (3.9B vs. 7B).
>
>
> Q2.
> > "Scalability Concerns: How does the model's performance and efficiency scale with longer videos, and have you observed any limitations in terms of the number of frames the model can effectively process?"
>
> Thanks for the suggestion. Since the number of frames our model takes is a hyperparameter, we are able to train a model that takes more frames as an input. In order to confirm that our model has a capability to digest a larger number of frames and still abstract each video into 32 (or 128) tokens, we trained BLIP-3-Video with 16 frames.
>
> The below table shows the trend.
>
> | # frames | # tokens | NExT-QA | ActivityNet-QA |
> | --- | --- | --- | --- |
> | 8 frames  | 32 tokens  | 76.4 | 55.7 / 3.5 |
> | 8 frames  | 128 tokens  | 77.1 | 56.7 / 3.6 |
> | 16 frames  | 32 tokens  |  76.7 | 55.9 / 3.5 |
> | 16 frames  | 128 tokens  | 77.6 | 57.3 / 3.6 |
>
> Even while maintaining the number of tokens, we are able to observe that providing more frames in the input allows BLIP-3-Video to scale to better performance. We believe this is due to the fact that increasing the number of frames has an effect of increasing the size of the "pool" of tokens the temporal encoder can select from. We believe this trend (i.e., our model accuracy increasing as the number of frames increases) will continue until it saturates.
>
> For very long videos (e.g., hours), it is beyond the scope of this paper, and it will require additional (hierarchical) LLM mechanisms similar to LLovi, LangRepo, or VideoTree. Such frameworks are complementary to ours, and we focus on best capturing temporal information over shorter (~2 min) video segments, which could be combined within such frameworks as also discussed a bit in the above post (Answer 2).
>
>
> Q3.
> > "Model Interpretability: The paper mentions the use of different types of temporal encoders. Are there any plans to provide insights into how these encoders make decisions?"
>
> Thanks for the suggestion. Our plan is to release the open source code with the final version of the paper. We will make the code include a visualization module for the attention layers, so that the user can check which visual tokens are being selected by which temporal encoder.

---

> ### Author Response · Authors · 2024-11-23
> **Response to reviewer 7c93 (4/4)**
>
> Q4.
> > "Comparison with Other Efficient Models: How does BLIP-3-Video compare with other recent models that also focus on efficiency, such as those employing knowledge distillation or sparse attention mechanisms? Could the authors provide some insights into the trade-offs involved? Please provide persuasive evidence from experimental studies."
>
> Knowledge distillation and sparse attention mechanisms certainly are interesting research directions to build an efficient, compact model. We believe such directions are orthogonal and complementary to ours. If there is an approach to distill a larger model into a smaller model or enable efficient attention, we believe our model could be extended to incorporate them. In this paper, what we focus on is the role of the temporal encoder and the observation that it is an effective component to abstract tokens over frames. For more detailed discussion, it will be very helpful to hear from the reviewer which efficient video-based VLM models the reviewer has in mind.
>
>
> Q5.
> > "Novelty: Compared to LLaMA-VID, where is the core novelty of this paper? Although the experimental results show that 32 tokens achieved better performance on four short video benchmarks, this standard will change with different video lengths, video scenarios, and the complexity of question answering. The scalability and generalizability of this method are questionable. Perhaps a more effective mechanism for accommodating more frames and selecting key information for video question answering from a large number of visual tokens is worth exploring, rather than determining the specific numerical value of a visually overfitted token on a few benchmarks. Similar architectures have been explored enough in a series of works such as Video-LLaVA, LLaVA-VID, LLaVA-NEXT, and so on."
>
> The key difference between LLaMA-VID and BLIP-3-Video’s token reduction is the existence of *temporal* encoder. LLaMA-VID token reduction is strictly per-frame; it only considers tokens within one frame to reduce them to 2 per frame, making it only spatial. On the other hand, our temporal encoder mechanism jointly looks at the tokens in the video across space and time. This allows BLIP-3-Video to more dynamically select informative tokens across all the frames. In an extreme case, BLIP-3-Video can learn to select 0 tokens from uninformative frames and 10+ tokens from another frame with important details.
>
> This contributes to the better accuracy of BLIP-3-Video compared to LLaMA-VID in all of the datasets we tried. The table below summarizes their comparisons.
>
> | Dataset | LLaMA-VID | BLIP-3-Video |
> | --- | --- | --- |
> | MSVD-QA | 70.0 / 3.7 | 77.1 / 4.2 |
> | MSRVTT-QA | 58.9 / 3.3 | 60.0 / 3.6 |
> | ActivityNet-QA | 47.5 / 3.3 | 55.7 / 3.5 |
> | VideoInstruct | 2.89 | 3.11 |
> | TempCompass (y/n) | 52.96 | 66.7 |
> | MVBench | 41.4 | 54.9 |
>
> Also notice that BLIP-3-Video (3.9B) achieves superior accuracy to LLaMA-VID (7B or 13B) while using a smaller LLM. We also tried an ablation against such a per-frame token reduction strategy (similar to LLaMA-VID) in Table 4, and obtained a similar observation. Our space-time token reduction enables much better performance.
>
> Another thing we would like to clarify is that the temporal encoder we are introducing in BLIP-3-Video itself has the capability to abstract tokens over any number of frames. Our *grouped Token Turing Machine* temporal encoder is a 'sequential model'. It in principle is able to sequentially (iteratively) digest tokens from a continuous stream of video frames. The current BLIP-3-Video’s design of taking a fixed number of frames (8 frames or 16 frames) originates from the limitation in our training hardware, but we believe we are showing the potential of the temporal encoder mechanism that could benefit future model designs. We believe it is much more generic than per-frame token reduction used in prior works like LLaMA-VID. Video-LLaVA and LLaVA-NEXT do not have any temporal encoder or token reduction mechanism over space-time either.

---

> ### Comment · Reviewer_7c93 · 2024-11-26
>
> Thank you to the authors for the detailed and well-reasoned responses, which have addressed some of my concerns. Consider incorporating recent advances in spatiotemporal token compression, such as LongVU, to enhance your architectural design instead of relying solely on the current framework. That said, I do not dismiss the substantial experimental contributions and technical robustness of this work. Overall, I find the insights this paper offers to video understanding researchers somewhat limited. Therefore, I am willing to increase my score from 5 to 6 but see no sufficient justification for a higher score.
>
> [1] Shen, Xiaoqian, et al. "LongVU: Spatiotemporal Adaptive Compression for Long Video-Language Understanding." arXiv preprint arXiv:2410.17434 (2024).

---

> > ### Author Response · Authors · 2024-11-26
> >
> > Thank you for bringing this very recent paper (LongVU) into our attention. We find that it was released on arXiv after we submitted our paper to ICLR.
> >
> > The frame selection used in LongVU certainly seems interesting and complementary to ours. The main difference between our approach and LongVU for the token compression is the temporal token abstract/merging over time. LongVU introduces a method to compress tokens in every frame spatially (i.e., within the same frame) while conditioning it on the first frame. On the other hand, our temporal encoder specializes in combining tokens over time (in all the frames), using the sequential model - grouped TTM.
> >
> > We believe these research works are complementary, focusing on different aspects of the problem. They will both benefit future research.

---

> > > ### Comment · Reviewer_7c93 · 2024-11-27
> > >
> > > Please ensure that all discussion details and additional experimental results are presented exactly as they are in the final paper, along with the complete code being open-sourced.

---

> > > > ### Author Response · Authors · 2024-11-28
> > > >
> > > > We plan to include the experimental results and discussions in the revised paper.
> > > >
> > > > Our submitted paper also mentions that the code will be open sourced together with the final version of the paper.

---

### Official Review · Reviewer_gZaZ · 2024-11-02

**Soundness:** 3
**Presentation:** 2
**Contribution:** 3
**Rating:** 6
**Confidence:** 4

**Summary:**

This paper presents an efficient Video LLM, coined as BLIP-3-Video, by incorporating extra modules to transform dense video tokens into sparse tokens (e.g., # can be 32). Specifically, the authors use a sequential transformer (so called token turing machine) on top of frame-wise image tokens and perceiver-resampler to produce limited 32 tokens. Diverse video QA benchmarks show the competitive performance of BLIP-3-Video on video question answering tasks. The authors also test the captioning ability of BLIP-3-Video.

**Strengths:**

This paper investigates an important topic, how to efficiently & effectively understand videos by LLMs, which is underexplored so far. This paper proposes the BLIP-3-Video to understand videos with just 32 tokens (in LLMs) based on Phi-3, and it outperforms both parameter-heavy or visual-token-heavy models (as shown in Fig 1) on QA and captioning benchmarks. Additionally, this paper presents a compelling finding (somewhat): a video can be effectively represented by just 32 tokens in LLMs for QA and captioning tasks. I think this research line is promising and can benefit several downstreaming tasks, e.g., captioning for text-to-video generation.

**Weaknesses:**

Although this is overall a good paper, several concerns are here:
1. The presentation of the main method (Sec 2.2) somwhat presents confusion: Does BLIP-3-Video both use spatio-temporal attentional pooling and TTM? Is there a perceive resampler before temporal encoder in BLIP-3-Video (cannot be infered from Figure 2)?

2. Compressing a video into 32 tokens is a compelling and exciting idea. However, I am worried that spatial-temporal details will be missing through compression, which is crucial for some detailed reasoning in LLMs. More evaluation of BLIP-3-Video on diverse tasks beyond captioning and MCQ are encouraged.
(also, as the compression is not text query guided, the compression is solely dominated by the visual information itself. That is to say, 32 tokens per a video are fixed under different text query, which might not be appropriate in general)

3. Relating with the weakness 1, extra modules introduced besides the visual encoder (SigLIP) and LLM sound too complicated. If I understand correctly, there are a perceiver-resampler and a temporal encoder (attention pooling or TTM). My idea is naive and simple, can we just finetune a perceiver-resampler in BLIP-3 into a temporal encoder, rather than just compressing tokens per frame? Given the strong performance of cross attention layers in the perceiver resampler, this seems to be a missing but promising ablation study in this paper.

**Questions:**

1. Can one BLIP-3-Video model produce 32 and 128 tokens on demand as a hyper parameter? Or they are different models trained individually?

2. What does text encoder in Fig 2 mean? The text tokenizer?

---

> ### Author Response · Authors · 2024-11-23
> **Response to reviewer gZaZ (1/2)**
>
> We thank the reviewer for the comments. Please find our answers below.
>
> > 1. "The presentation of the main method (Sec 2.2) somwhat presents confusion: Does BLIP-3-Video both use spatio-temporal attentional pooling and TTM? Is there a perceive resampler before temporal encoder in BLIP-3-Video (cannot be infered from Figure 2)?"
>
> We apologize for the confusion. We have two different versions of the model architecture: one that uses spatio-temporal attentional pooling, and the other that uses TTM as the temporal encoder.
>
> For the TTM version, we clarify that the TTM itself has the final “output” operation layer in it (by its design), which we implement using a TokenLearner (spatio-temporal attentional pooling). As a result, the TTM encoder naturally contains spatio-temporal attentional pooling within it. We will improve the figure to clarify this better, and thank you for the suggestion.
>
> In both the versions, yes, Perceiver-Resampler exists before the temporal encoder. We are inheriting it from the BLIP-3 model. Perceiver-Resampler generates 128 tokens per frame, and our temporal encoder is applied on top of it to map such 128 * T tokens into 32 (or 128) tokens total. We will revise Figure 2 to clarify this further.
>
> > 2. "Compressing a video into 32 tokens is a compelling and exciting idea. However, I am worried that spatial-temporal details will be missing through compression, which is crucial for some detailed reasoning in LLMs. More evaluation of BLIP-3-Video on diverse tasks beyond captioning and MCQ are encouraged. (also, as the compression is not text query guided, the compression is solely dominated by the visual information itself. That is to say, 32 tokens per a video are fixed under different text query, which might not be appropriate in general)"
>
> Yes, we do understand the concern and we thank the reviewer for asking this.
>
> The focus of this paper is on the learning of compact *visual* representation prior to their interaction with text. It will have pros and cons. One advantage would be that this allows answering multiple questions without having to re-compute the visual tokens.
>
> Following the suggestion regarding the diverse tasks, we further evaluated BLIP-3-Video on TempCompass benchmark, in order to test it on more various tasks other than video captioning and MCQ, TempCompass is particularly useful as it has two different types of evaluations in addition to MCQ and captioning: "yes/no QA" and "caption matching". We tested BLIP-3-Video on these two tasks. Also notice that TempCompass has some explicit temporal reasoning questions, such as the “event order” and “speed” questions in the dataset.
>
> We find BLIP-3-Video quite competitive in this benchmark, particularly considering its size (3.9B) and the number of visual tokens (often less than 1/20 of the others). Notably, we performed quite a bit better than LLaMA-VID-7B, which also uses 32 visual tokens like ours (and has a larger LLM).
>
> | Model | Yes/No QA | Caption matching |
> | --- | --- | --- |
> | GPT-4o | 73.66 | 80.84 |
> | Qwen2-VL-7B-Instruct | 72.77 | 77.31 |
> | Gemini-1.5-pro | 70.32 | 77.45 |
> | LLaVA-OneVision-Qwen-2-7B | 69.67 | 73.79 |
> | LLaVA-NeXT-Video-32B-Qwen | 69.38 | 76.51 |
> | InternVL2-8B | 68.24 | 77.11 |
> | **BLIP-3-Video (3.9B)** | 66.7 | 66.5 |
> | Llama-3-VILA1.5-8B | 63.64 | 68.93 |
> | LongVA-7B | 62.13 | 65.67 |
> | LLaVA-NeXT-Video-7B-DPO | 61.19 | 63.01 |
> | VideoChat2-vicuna-stage3 | 58.01 | 53.69 |
> | LLaVA-1.5-13B | 56.38 | 64.27 |
> | Video-LLaVA-7B | 56.38 | 63.34 | 63.34 |
> | Video-LLaMA-2-13B | 53.73 | 54.16 |
> | LLaMA-VID-7B-short-video | 52.96 | 56.02 |
>
> We also tested BLIP-3-Video on another benchmark, MVBench, and got a similar observation.
>
> | Model | # tokens | {VideoChat2-IT training} | Accuracy |
> | --- | --- | :---: | --- |
> | PLLaVA (7B) | 576+ | Y | 46.6 |
> | VideoLLaMA2 (7B) | 1152 | Y | 54.6 |
> | ST-LLM (7B) | 256 | ~Y | 54.9 |
> | PPLLaVA (7B) | 1024 | ~Y | 59.2 |
> | VideoChat2-Mistral (7B) | 96 | Y | 60.4 |
> | Kangaroo (8B) | ~10000 | Y | 61.1 |
> | Tarsier (7B) | 4608+ | ~Y | 62.6 |
> | | | | |
> | VideoChatGPT (7B) | 264+ | N | 32.7 |
> | VideoLLaMA (7B) | 32 | N | 34.1 |
> | VideoChat (7B) | 32 | N | 35.5 |
> | LLaMA-VID (7B) | 32 | N | 41.4 |
> | Video-LLaVA (7B) | 2048 | N | 43.5 |
> | mPLUG-Owl3 (8B)| n/a | N | 54.5 |
> | **BLIP-3-Video (3.9B)** | 32 | N | 54.9 |
> | LLaVA-OneVision (7B) | 3136 | N | 56.7 |
>
> Also note that NExT-QA, which we already included in the submitted paper, has some explicit temporal ordering questions.

---

> ### Author Response · Authors · 2024-11-23
> **Response to reviewer gZaZ (2/2)**
>
> > 3. "Relating with the weakness 1, extra modules introduced besides the visual encoder (SigLIP) and LLM sound too complicated. If I understand correctly, there are a perceiver-resampler and a temporal encoder (attention pooling or TTM). My idea is naive and simple, can we just finetune a perceiver-resampler in BLIP-3 into a temporal encoder, rather than just compressing tokens per frame? Given the strong performance of cross attention layers in the perceiver resampler, this seems to be a missing but promising ablation study in this paper."
>
> We agree that Perceiver-Resampler adds one more layer of complication. Its existence is solely because it is part of the pre-trained BLIP-3 model we build on top of, and we are directly inheriting it. The concept of temporal encoder we demonstrate in the paper would be independent of the existence of the Perceiver-Resampler.
>
> Following the suggestion from the reviewer, we ran an additional ablation. We implemented the model only using/finetuning Perceiver-Resampler in BLIP-3, making it also serve as the temporal encoder. In this version, the Perceiver-Resampler compresses tokens across all the frames, and there is no separate spatio-temporal pooling or TTM.
>
> The table below shows the results. 128 tokens were used in this table for a fair comparison, as the Perceiver-Resampler was pre-trained and fine-tuned to extract 128 tokens. We observe that fine-tuning Perceiver-Resampler as a temporal encoder provides much worse performance compared to our temporal encoders.
>
> | Temporal encoder | TGIF-QA | ActivityNet-QA  | NExT-QA |
> | ------------------------------ | ----------- | ----------- | --------- |
> | Perceiver-Resampler | 72.46 / 4.13 | 52.61 / 3.38 | 76.44 |
> | Ours (Attentional Pooling)        | 76.90 / 4.29   |  56.94 / 3.56  | 76.27 |
> | Ours (Grouped TTM)        |  77.10 / 4.31   |  56.66 / 3.56 | 77.07 |
>
>
> > Q1. "Can one BLIP-3-Video model produce 32 and 128 tokens on demand as a hyper parameter? Or they are different models trained individually?"
>
> We do need to train a different model separately if the number of target visual tokens changes. More dynamic selection with a single model will be an interesting future work. We will clarify this in the paper.
>
> > Q2. "What does text encoder in Fig 2 mean? The text tokenizer?"
>
> Yes, it’s the text tokenizer. Apologizes for the confusion, and we will revise it.

---

> > ### Comment · Reviewer_gZaZ · 2024-11-25
> >
> > Thank you for efforts and detailed rebuttal and all my concerns are comprehensively addressed. I have carefully read the author's response as well as the feedback from other reviewers and I will maintain my borderline acceptance score as it is.
> >
> > It would be great that the authors to investigate the variant with variable visual token numbers on demand in the future version, as mentioned in Q1. In my view attentional pooling or grouped ttm can naturally manage variable video tokens.

---

> > > ### Author Response · Authors · 2024-11-26
> > >
> > > We thank the reviewer for the feedback.
> > >
> > > We are very glad to know that all of the concerns have been comprehensively addressed. The suggestion for a new model with dynamic token number is a great idea worth exploring in the future, and we will certainly include its discussion in the final version of the paper.
> > >
> > > Please let us know if there are any further things we can provide or clarify. We are a bit lost as the score remains borderline acceptance even after addressing all the concerns.

---

### Official Review · Reviewer_hUbh · 2024-11-04

**Soundness:** 3
**Presentation:** 3
**Contribution:** 4
**Rating:** 8
**Confidence:** 4

**Summary:**

The paper introduces BLIP-3-Video, a multimodal vision language model which demonstrates strong performance on video understanding with high token accuracy. BLIP-3-Video uses as few as 16~32 tokens to encode an entire video sequence, which is highly efficient to other video VLMs, enabled by the incorporation of a temporal encoder. BLIP-3-Video achieves competitive accuracy on various video question-answering benchmarks while having smaller model parameters.

**Strengths:**

- The temporal encoder which aggregates visual tokens across frames in a highly efficient manner results in computational efficiency of training the model.
- Extensive ablation study in the temporal encoder design validates the design choice while also demonstrating the flexibility in the design of temporal encoder.
- Competitive performance on various video question-answering benchmarks, despite its smaller size.

**Weaknesses:**

- The model proposed in the paper utilizes 8 frames per video which are uniformly sampled. This approach might not work for tasks that inherently require more than 8 frames to understand the video. If this method could scale up, an explanation of why that might be would be helpful.
- The experiments of the paper focuses on video question-answering benchmarks only, and this limited experimentation may not capture the model's ability in other video-based tasks. Further evaluation on other video tasks, such as temporal understanding would demonstrate the applicability of this approach to more general and diverse video-related tasks.

**Questions:**

- The (video part of the) training of this model is on video captioning data and video question-answering datasets. If the downstream task were to change to a more complex task, like temporal reasoning, would the model require more tokens or would 16~32 still be sufficient? i.e. is there enough visual information encoded in the 16~32 tokens?
- In addition, if the downstream task requires remembering multiple details and nuanced events over a long diverse scenario, how would this approach perform? Is there a built-in mechanism that prevents information loss during token pooling?

---

> ### Author Response · Authors · 2024-11-23
> **Response to reviewer hUbh (1/3)**
>
> We thank the reviewer for the comments. Please find our answers below.
>
> > 1. “The model proposed in the paper utilizes 8 frames per video which are uniformly sampled. This approach might not work for tasks that inherently require more than 8 frames to understand the video. If this method could scale up, an explanation of why that might be would be helpful.”
>
> We thank the reviewer for raising the concern. The number of frames our model takes is a hyperparameter, and we are able to train a model that takes more frames as an input when necessary. In order to confirm that our model has a capability to digest a larger number of frames and still abstract each video into 32 (or 128) tokens, we trained BLIP-3-Video with 16 frames.
>
> The below table shows the trend.
>
> | # frames | # tokens | NExT-QA | ActivityNet-QA |
> | --- | --- | --- | --- |
> | 8 frames  | 32 tokens  | 76.4 | 55.7 / 3.5 |
> | 8 frames  | 128 tokens  | 77.1 | 56.7 / 3.6 |
> | 16 frames  | 32 tokens  | 76.7 | 55.9 / 3.5 |
> | 16 frames  | 128 tokens  | 77.6 | 57.3 / 3.6 |
>
> Even while maintaining the number of tokens, we are able to observe that providing more frames in the input allows BLIP-3-Video to scale to better performance. We believe this is due to the fact that increasing the number of frames has an effect of increasing the size of the "pool" of tokens the temporal encoder can select from. We believe this trend (i.e., our model accuracy increasing as the number of frames increases) will continue until it saturates.
>
> We have not observed much sign of information loss with the datasets we tried, including ActivityNet-QA whose average duration is 111 seconds and NExT-QA whose duration is 44 seconds. What we confirm in this paper is that our proposed architecture with temporal encoder could be a useful concept/component for properly representing video clips of such durations.
>
> Another thing we would like to mention is that our model is capable of handling any frames provided as an input (i.e., they don’t need to be uniformly sampled). Frame selection research would be orthogonal and complementary to our work, and they can be easily combined within BLIP-3-Video. The focus of this paper is on confirming the potential of the temporal encoder to capture necessary information in the given frames (and what frames to give is complementary). If a frame selection algorithm is incorporated, we expect its even better scaling capability.

---

> ### Author Response · Authors · 2024-11-23
> **Response to reviewer hUbh (2/3)**
>
> > 2. "The experiments of the paper focus on video question-answering benchmarks only, and this limited experimentation may not capture the model's ability in other video-based tasks. Further evaluation on other video tasks, such as temporal understanding would demonstrate the applicability of this approach to more general and diverse video-related tasks."
>
> We thank the reviewer for the suggestion. We also want to mention that we evaluated the model on a video caption task (in addition to VQA), and reported the results in Section 3.4.
>
> In addition, we newly evaluated BLIP-3-Video on TempCompass, in order to test it on more various tasks other than video captioning and MCQ, TempCompass is particularly useful as it has two different types of evaluations in addition to MCQ and captioning: "yes/no QA" and "caption matching". We tested BLIP-3-Video on these two tasks. Also notice that TempCompass has some explicit temporal reasoning questions, such as the “event order” and “speed” questions in the dataset.
>
> We find BLIP-3-Video quite competitive in this benchmark, particularly considering its size (3.9B) and the number of visual tokens (often less than 1/20 of the others). Notably, we performed quite a bit better than LLaMA-VID-7B, which also uses 32 visual tokens like ours (and has a bigger LLM).
>
> | Model | Yes/No QA | Caption matching |
> | --- | --- | --- |
> | GPT-4o | 73.66 | 80.84 |
> | Qwen2-VL-7B-Instruct | 72.77 | 77.31 |
> | Gemini-1.5-pro | 70.32 | 77.45 |
> | LLaVA-OneVision-Qwen-2-7B | 69.67 | 73.79 |
> | LLaVA-NeXT-Video-32B-Qwen | 69.38 | 76.51 |
> | InternVL2-8B | 68.24 | 77.11 |
> | **BLIP-3-Video (3.9B)** | 66.7 | 66.5 |
> | Llama-3-VILA1.5-8B | 63.64 | 68.93 |
> | LongVA-7B | 62.13 | 65.67 |
> | LLaVA-NeXT-Video-7B-DPO | 61.19 | 63.01 |
> | VideoChat2-vicuna-stage3 | 58.01 | 53.69 |
> | LLaVA-1.5-13B | 56.38 | 64.27 |
> | Video-LLaVA-7B | 56.38 | 63.34 | 63.34 |
> | Video-LLaMA-2-13B | 53.73 | 54.16 |
> | LLaMA-VID-7B-short-video | 52.96 | 56.02 |
>
>
> We also tested BLIP-3-Video on another benchmark, MVBench, and got a similar observation.
>
> | Model | # tokens | {VideoChat2-IT training} | Accuracy |
> | --- | --- | :---: | --- |
> | PLLaVA (7B) | 576+ | Y | 46.6 |
> | VideoLLaMA2 (7B) | 1152 | Y | 54.6 |
> | ST-LLM (7B) | 256 | ~Y | 54.9 |
> | PPLLaVA (7B) | 1024 | ~Y | 59.2 |
> | VideoChat2-Mistral (7B) | 96 | Y | 60.4 |
> | Kangaroo (8B) | ~10000 | Y | 61.1 |
> | Tarsier (7B) | 4608+ | ~Y | 62.6 |
> | | | | |
> | VideoChatGPT (7B) | 264+ | N | 32.7 |
> | VideoLLaMA (7B) | 32 | N | 34.1 |
> | VideoChat (7B) | 32 | N | 35.5 |
> | LLaMA-VID (7B) | 32 | N | 41.4 |
> | Video-LLaVA (7B) | 2048 | N | 43.5 |
> | mPLUG-Owl3 (8B)| n/a | N | 54.5 |
> | **BLIP-3-Video (3.9B)** | 32 | N | 54.9 |
> | LLaVA-OneVision (7B) | 3136 | N | 56.7 |

---

> ### Author Response · Authors · 2024-11-23
> **Response to reviewer hUbh (3/3)**
>
> > Q1. "The (video part of the) training of this model is on video captioning data and video question-answering datasets. If the downstream task were to change to a more complex task, like temporal reasoning, would the model require more tokens or would 16~32 still be sufficient? i.e. is there enough visual information encoded in the 16-32 tokens?"
>
> We agree that whether 32 tokens would be sufficient is an open question. There is a good chance that very long and complicated videos may require more frames and tokens. However, we would like to clarify that our observation from this paper is expected to still hold: the current video-based VLM models (without a proper temporal encoder) have too many visual tokens, and (intelligently) reducing them to 1/20 does not harm the accuracy and only makes it more efficient. For more complicated tasks requiring more frames, we believe all of the models will need to increase the number of tokens for the best accuracy. Simultaneously, BLIP-3-Video is likely to always require less number of tokens compared to the naive video VLMs without the temporal encoder, relatively. For the benchmarks we tested, we were able to confirm such behavior.
>
>
>
> > Q2. "In addition, if the downstream task requires remembering multiple details and nuanced events over a long diverse scenario, how would this approach perform? Is there a built-in mechanism that prevents information loss during token pooling?"
>
> Yes, our temporal encoder (e.g., TTM) has the 'memory' mechanism in it, and it is expected to learn to select/preserve important tokens that best benefit the training task. If the training data with such property is provided, the temporal encoder will be optimized that way. Obviously, if there are too many important tokens, it will reach its capacity. (The size of the memory in TTM was 512 tokens, which is before they are finally pooled to 32 or 128.)
>
> The focus of this paper has been on capturing temporal information over shorter (~2 min) video segments. What we show in the paper is that our model with the temporal encoder mechanism enables efficient capturing of information in video segments compared to prior work. What we claim is that it is a better/more efficient way to handle a video clip.
>
> The very long videos are outside the scope of this research work. However, this is an important research area, and we can offer the following insights. In order to handle long videos (e.g., movies), additional (hierarchical) LLM mechanisms similar to LLovi, LangRepo, or VideoTree would be very useful. Such framework design is orthogonal to our work; our model can serve as the VLM component within those types of frameworks, enabling extraction of better video information from local/short segments that higher-level LLM will combine for the long video modeling.
>
> We will clarify and discuss this further in the final version of the paper.

---

> ### Comment · Reviewer_hUbh · 2024-11-25
> **Great rebuttal**
>
> I thank the authors for their detailed rebuttal. I have carefully read other reviews and the rebuttals. I appreciate their detailed response and additional experiments.
>
> To clarify, by no means was I trying to disregard their contributions/observations in the paper, but I felt the paper would become much stronger if they could provide their insights into some of the points I mentioned (e.g. number of frames, frame selection, complex motions, etc.)
>
> I believe that the insights the authors provide will be a valuable contribution to the community, and that BLIP-3 will be a promising direction of future research, given their efficiency in parameter utilization.
>
> Hence I increase my score and confidence accordingly.

---

> > ### Author Response · Authors · 2024-11-25
> >
> > We thank the reviewer very much for the constructive feedback and understanding the contributions of this work.

---

### Official Review · Reviewer_yYZF · 2024-11-10

**Soundness:** 2
**Presentation:** 2
**Contribution:** 2
**Rating:** 5
**Confidence:** 4

**Summary:**

This paper presents BLIP-3-Video, which introduces a "temporal encoder" alongside a conventional visual tokenizer, allowing it to significantly reduce visual tokens (32 tokens compared to thousands in other models). The study explored various temporal encoders, including learnable spatio-temporal pooling and sequential models like token turning machines (TTM). Detailed experiments showed that different encoder types had a noticeable impact on performance, particularly in handling complex video scenarios. Experimental results show that BLIP-3-Video achieved video question-answering accuracies comparable to much larger state-of-the-art models while being smaller and more efficient.

**Strengths:**

1. An impressive tradeoff between efficiency and accuracy on the MSVD-QA benchmark.
2. Extensive explorations on temporal encoders to reduce visual tokens.
3. The paper is well written and easy to follow.

**Weaknesses:**

1. Novelty: The primary weakness is the insufficient novelty. As detailed in Section 2.2, the only improvements to TTM include (1) time-stamped positional encodings and (2) a 'grouped' TTM temporal encoder. These minor changes do not substantiate a significant contribution.

2. Evaluation Benchmarks: The evaluated benchmarks are unconvincing for assessing Video LMMs. The model was only evaluated on MSVD-QA, MSRVTT-QA, ActivityNet-QA, TGIF-QA, and NExT-QA, which are not so ideal for testing LMMs. The authors may consider newer benchmarks like VideoMME and MVBench, which are proposed for assessing Video LMMs.

**Questions:**

1. What novel designs does this method introduce compared to TTM? Are there ablation studies for these designs?

2. The model utilizes the VideoChatGPT instruction set. Why hasn't it been evaluated on that benchmark?

---

> ### Author Response · Authors · 2024-11-23
> **Response to reviewer yYZF (1/3)**
>
> We thank the reviewer for the comments. Please find our answers to the reviewer's comments below.
>
> > 1. "Novelty: The primary weakness is the insufficient novelty. As detailed in Section 2.2, the only improvements to TTM include (1) time-stamped positional encodings and (2) a 'grouped' TTM temporal encoder. These minor changes do not substantiate a significant contribution."
>
> Although TTM was introduced in the previous work, we find our extensions very crucial and impactful, particularly for video-based VLMs. We observe that the original TTM (as it is) does not perform competitively within the VLM, and that our extension improves this significantly.
>
> Please find the ablation result table comparing different extensions of the TTM below:
>
> | Temporal encoder | MSVD-QA | TGIF-QA | ActivityNet-QA  | NExT-QA |
> |-----------------------------------------------|-----------|-----------|---------|----------|
> | Original TTM         |  76.42 / 4.15   | 75.80 / 4.26    |  54.45 / 3.48 | 75.42 |
> | TTM + time-stamp        |  76.43 / 4.16   | 76.44 / 4.29    | 56.15 / 3.53  | 75.96 |
> | TTM + grouping         |  76.99 / 4.17  |  77.05 / 4.30   | 55.92 / 3.54   | 76.46  |
> | Ours (time-stamp + grouping)         |  77.29 / 4.18   | 77.10 / 4.31   |  56.66 / 3.56 | 77.07 |
>
> What we introduce in this paper is the extended TTM that actually works well within a multimodal LLM, unlike previous work.
>
> In addition, we believe that our paper is the first paper to successfully extend TTM for this significant token reduction within a video-based VLM. TTM was originally used with a much smaller Transformer (ViT), and its token reduction was only down to 16 per frame/timestep. Mirasol3B tried TTM within a VLM, but even in Mirasol3B, the tokens were reduced to 32 per timestep. In our case, the tokens are reduced to 2-4 per timestep (i.e., 16-32 tokens total).
>
> Finally, we would like to emphasize once more that most of the video-based VLMs omit the usage of temporal encoders entirely. Compared to these prior work without any temporal encoder (e.g., Tarsier, LLaVA-OneVision, …), we believe we are providing an efficient and effective architecture to summarize tokens over time. We are exploring the use of temporal encoders, a sequential model in particular, at this scale almost for the first time.
>
> We also emphasize that BLIP-3-Video is one of the first compact video models (3.9B) which obtains competitive performance to much larger SOTA models. We believe the findings of this paper (e.g., token reduction with temporal encoders) can further benefit future model designs in the research community.

---

> ### Author Response · Authors · 2024-11-23
> **Response to reviewer yYZF (2/3)**
>
> > 2. “Evaluation Benchmarks: The evaluated benchmarks are unconvincing for assessing Video LMMs. The model was only evaluated on MSVD-QA, MSRVTT-QA, ActivityNet-QA, TGIF-QA, and NExT-QA, which are not so ideal for testing LMMs. The authors may consider newer benchmarks like VideoMME and MVBench, which are proposed for assessing Video LMMs.”
>
> Following the suggestion from the reviewer, we evaluated our model on MVBench. The below table compares BLIP-3-Video with other state-of-the-art (SOTA) models on MVBench.
>
> | Model | # tokens | {VideoChat2-IT training} | Accuracy |
> | --- | --- | :---: | --- |
> | PLLaVA (7B) | 576+ | Y | 46.6 |
> | VideoLLaMA2 (7B) | 1152 | Y | 54.6 |
> | ST-LLM (7B) | 256 | ~Y | 54.9 |
> | PPLLaVA (7B) | 1024 | ~Y | 59.2 |
> | VideoChat2-Mistral (7B) | 96 | Y | 60.4 |
> | Kangaroo (8B) | ~10000 | Y | 61.1 |
> | Tarsier (7B) | 4608+ | ~Y | 62.6 |
> | | | | |
> | VideoChatGPT (7B) | 264+ | N | 32.7 |
> | VideoLLaMA (7B) | 32 | N | 34.1 |
> | VideoChat (7B) | 32 | N | 35.5 |
> | LLaMA-VID (7B) | 32 | N | 41.4 |
> | Video-LLaVA (7B) | 2048 | N | 43.5 |
> | mPLUG-Owl3 (8B)| n/a | N | 54.5 |
> | **BLIP-3-Video (3.9B)** | 32 | N | 54.9 |
> | LLaVA-OneVision (7B) | 3136 | N | 56.7 |
>
>
> Among the models which did not directly use MVBench-provided training dataset (i.e., VideoChat2-IT), BLIP-3-Video performs the 2nd only after LLaVA-OneVision which is bigger and uses almost ~100x more visual tokens. BLIP-3-Video achieves decent results given its smaller model size (3.9B vs. 7B+) and fewer visual tokens (32 vs. 3000+).
>
> We also hypothesize that training BLIP-3-Video with VideoChat2-IT, which is the instruction tuning data introduced by the MVBench paper, would further enhance its performance, since VideoChat2-IT contains datasets like CLEVRER which also exists in MVBench; many SOTA models achieving high results on MVBench did train their models using VideoChat2-IT.  In the table, ‘Y’ denotes that the model was trained on VideoChat2-IT, whereas ‘N’ denotes that the model was not trained on VideoChat2-IT. ‘~Y’ means the model’s paper does not explicitly say they used VideoChat2-IT, but their training recipe shows the majority of VideoChat2-IT was actually used (e.g., CLEVRER, Kinetics-710 & SthSthV2, WebVid, EgoQA, YouCook2, etc.).

---

> ### Author Response · Authors · 2024-11-23
> **Response to reviewer yYZF (3/3)**
>
> > Q1. "What novel designs does this method introduce compared to TTM? Are there ablation studies for these designs?"
>
> The table below (also mentioned in the above post) shows the ablations comparing different extensions of the TTM. As we are able to observe, the original TTM performs poorly, and the time-stamped positional encoding and formulating the grouped version enables much better results.
>
> | Temporal encoder | MSVD-QA | TGIF-QA | ActivityNet-QA  | NExT-QA |
> |-----------------------------------------------|-----------|-----------|---------|----------|
> | Original TTM         |  76.42 / 4.15   | 75.80 / 4.26    |  54.45 / 3.48 | 75.42 |
> | TTM + time-stamp        |  76.43 / 4.16   | 76.44 / 4.29    | 56.15 / 3.53  | 75.96 |
> | TTM + grouping         |  76.99 / 4.17  |  77.05 / 4.30   | 55.92 / 3.54   | 76.46  |
> | Ours (time-stamp + grouping)         |  77.29 / 4.18   | 77.10 / 4.31   |  56.66 / 3.56 | 77.07 |
>
>
>
> > Q2. "The model utilizes the VideoChatGPT instruction set. Why hasn't it been evaluated on that benchmark?"
>
> Following the suggestion from the reviewer, we evaluated our model on the VideoInstruct benchmark.
>
> | Model  | VideoInstruct accuracy |
> | --- | --- |
> | PLLaVA-34B | 3.32 |
> | SlowFast-LLaVA-34B | 3.32 |
> | VideoGPT+ | 3.28 |
> | ST-LLM-7B | 3.15 |
> | **BLIP-3-Video (3.9B)** | 3.11 |
> | VideoChat2_HD_mistral | 3.10 |
> | LITA-13B | 3.04 |
> | LLaMA-VID-13B | 2.99 |
> | VideoChat2 | 2.98 |
> | LLaMA-VID-7B | 2.89 |
> | Video-ChatGPT | 2.38 |
>
> It shows a similar trend to the experiments with other datasets. We believe BLIP-3-Video performs very reasonably on this benchmark, considering its smaller size and its use of much fewer visual tokens. Notably, we performed quite a bit better than LLaMA-VID-7B, which also uses 32 visual tokens like ours (and has a bigger LLM).

---

> > ### Comment · Reviewer_yYZF · 2024-11-25
> >
> > Thanks for the rebuttal. The performance on the VideoMME and VideoChatGPT benchmarks looks good. I will increase my rating. However, like reviewer 7c93, my concern about the novelty is not fully resolved. It is hard to identify "+grouping, +timestamp" as novelties; they seem incremental.

---

> > > ### Author Response · Authors · 2024-11-26
> > >
> > > We are glad to know that all the concerns have been resolved except about the novelty.
> > >
> > > As Reviewer hUbh also mentioned, we believe we have meaningful contributions and important observations (impact of token reduction, role of temporal encoder, ...) to share with the research community. We believe this will benefit and motivate future model designs by other researchers. We hope the reviewer will consider them when increasing the rating.

---

### Public Comment · ~Michael_S_Ryoo1 · 2025-02-13
**Post-decision comment**

This paper, which got 8(accept)-6(borderline accept)-6(borderline accept)-5(borderline reject) scores got rejected.

Although we respect the decision of the AC, we feel a bit sad that the paper is rejected despite having strong technical contributions to the field (supported by the reviewers), because of one remaining under-explained opinion of just a single reviewer.

It is particularly so, if all 4 reviewers agreed that the paper has meaningful technical observations by introducing a small video-VLM model with an extremely small number of visual tokens. Further, 3 out of 4 reviewers agreed that the paper presents important technical contributions to the research area introducing the extensive use of temporal encoders for visual tokens in the video. Notice that, as mentioned in our interactions with the reviewers and acknowledged by them, most of the existing video-based VLMs entirely lack the use of temporal encoders on top of image-level visual encoders, setting aside extending advanced sequential models like Token Turning Machines like what we do in this paper.

We emphasize once more that all the concerns raised by the reviewers (mentioned in the AC comment) were addressed during our rebuttal, except for the difference in opinion with yYZF, which was given to us without much justification or details.

---

### Meta-Review · Area_Chair_ETK1 · 2024-12-22

**Metareview:**

This paper provides BLIP-3-Video, a Video LLM that build on the previous BLIP-3 architecture but focuses on incorporating temporality into the architecture by learning spatio-temporal pooling to obtain video representations in only 32 tokens. The paper is clearly written, provides a thorough analysis of various pooling strategies and tackles a topic of high interest to the community.
The paper's key weakness is the limited novelty, it is rather an exploration of architectures. Especially compared to TTM, the novelty, as also stated by the authors is only the addition of grouping and time-stamp. For this reason, the AC recommends the rejection of the paper as its key contributions don't warrant a full paper at ICLR.

**Additional Comments On Reviewer Discussion:**

The comment to the AC has been considered. Overall reviewers raised points about novelty (yYZF, 7c93), scalability questions (7c93), flexibility in output tokens after training (gZaZ), and further evaluations on other tasks, including ones that require longer memory (hUbh). These points were discussed in a thorough set of discussions with the authors, and mainly points about novelty remaining. It is a difficult borderline paper and the key to the decision by the AC has been the overall lack of core technical contribution that would warrant a full ICLR paper.

---

### Decision · Program_Chairs · 2025-01-22

Reject